# Uremic Toxins and Ciprofloxacin Affect Human Tenocytes In Vitro

**DOI:** 10.3390/ijms21124241

**Published:** 2020-06-14

**Authors:** Erman Popowski, Benjamin Kohl, Tobias Schneider, Joachim Jankowski, Gundula Schulze-Tanzil

**Affiliations:** 1Department of Traumatology and Reconstructive Surgery, Campus Benjamin Franklin, Charité–Universitätsmedizin Berlin, Freie Universität Berlin, Humboldt-Universität zu Berlin and Berlin Institute of Health, Hindenburgdamm 30, 12203 Berlin, Germany; e.popowski@web.de (E.P.); benjamin.kohl@charite.de (B.K.); tob.schn@t-online.de (T.S.); 2Institute of Anatomy, Paracelsus Private Medical University, Nuremberg and Salzburg, Nuremberg, Prof. Ernst Nathan Str. 1, 90419 Nuremberg, Germany; 3Institute for Molecular Cardiovascular Research (IMCAR), RWTH Aachen, Pauwelsstraße 30, 52074 Aachen, Germany; jjankowski@ukaachen.de; 4Experimental Vascular Pathology, Cardiovascular Research Institute Maastricht (CARIM), Maastricht University, 6229 HX Maastricht, The Netherlands

**Keywords:** ciprofloxacin, tenocytes, uremic toxins, phenylacetic acid, quinolinic acid, integrin, interleukin-1beta, matrix metalloproteinase-1

## Abstract

Tendinopathy is a rare but serious complication of quinolone therapy. Risk factors associated with quinolone-induced tendon disorders include chronic kidney disease accompanied by the accumulation of uremic toxins. Hence, the present study explored the effects of the representative uremic toxins phenylacetic acid (PAA) and quinolinic acid (QA), both alone and in combination with ciprofloxacin (CPX), on human tenocytes in vitro. Tenocytes incubated with uremic toxins +/- CPX were investigated for metabolic activity, vitality, expression of the dominant extracellular tendon matrix (ECM) protein type I collagen, cell-matrix receptor β1-integrin, proinflammatory interleukin (IL)-1β, and the ECM-degrading enzyme matrix metalloproteinase (MMP)-1. CPX, when administered at high concentrations (100 mM), suppressed tenocyte metabolism after 8 h exposure and at therapeutic concentrations after 72 h exposure. PAA reduced tenocyte metabolism only after 72 h exposure to very high doses and when combined with CPX. QA, when administered alone, led to scarcely any cytotoxic effect. Combinations of CPX with PAA or QA did not cause greater cytotoxicity than incubation with CPX alone. Gene expression of the pro-inflammatory cytokine IL-1β was reduced by CPX but up-regulated by PAA and QA. Protein levels of type I collagen decreased in response to high CPX doses, whereas PAA and QA did not affect its synthesis significantly. MMP-1 mRNA levels were increased by CPX. This effect became more pronounced in the form of a synergism following exposure to a combination of CPX and PAA. CPX was more tenotoxic than the uremic toxins PAA and QA, which showed only distinct suppressive effects.

## 1. Introduction

The first report concerning the tenotoxicity of quinolone antibiotics was published decades ago [1]. Ciprofloxacin (CPX) is one of the most common drugs associated with this adverse effect [2,3] and has been implicated in quinolone-associated ruptures of various tendons, such as Achilles tendons or those of the triceps brachii, iliopsoas, gluteal, adductor longus, and extensor digitorum communis muscles [2,4,5,6,7,8,9,10,11,12,13].

Why tendons are particularly prone to the toxic side effects of fluoroquinolones remains unclear. Tendons are a bradytrophic tissue with poor blood supply and low cell content [14]. Most of the cells within tendons are tenocytes, representing specialized fibroblasts. It is likely that the low nutrient exchange exposes tendons to the toxic effects of this class of antibiotics. However, the particular signaling pathways involved in mediating quinolone-driven tenotoxicity, as well as mechanisms of synergistic and additive effects with other therapeutic agents and metabolic conditions, have not been elucidated thus far.

Several hypotheses (Figure 1) have been proposed based on experimental observations associated with quinolones in tendon, such as inhibition of tendon-derived cell proliferation, occurrence of cell cycle arrest, impaired tenocyte migration, and dysregulated ECM synthesis, for example, elevated type III collagen expression (increase in the type III/I collagen ratio) [15,16,17,18,19,20]. In addition, reduced cell viability, impaired expression of the divalent cation-dependent β1-integrin receptors, N-cadherins, and connexins; increased production of several matrix metalloproteinases (MMPs) [21,22,23,24,25] coupled with extracellular tendon matrix (ECM) degradation and oxidative stress [25,26,27] have previously been implicated in tenotoxicity of quinolones (Figure 1). MRI analysis revealed a loss in glycosaminoglycan (GAG) content induced by CPX [28]. In recent years, some predispositions to quinolones-mediated tendinopathies have been documented, such as treatment with glucocorticoids [29], old age, athletes, and renal insufficiency or transplantation [2,12,30]. In particular, renal insufficiency leads to metabolic dysregulations such as secondary hyperparathyroidism and hypercalcinosis, facilitating tendon ruptures [31,32,33,34,35,36]. However, renal insufficiency is complex and goes along with the increase of several toxic compounds, so-called uremic toxins, which cannot be fully removed by hemodialysis [37,38,39]. Moreover, CPX is predominately renally eliminated [40], hence, variable serum concentrations could be expected in patients with chronical kidney diseases and possibly, also an accumulation in the bradytrophic tendon tissue.

Some of these uremic toxins, such as phenylacetic acid (PAA), are partly protein bound, by, for example, binding albumin [41]. PAA is responsible for activating several functions of polymorphonuclear leukocytes, accompanied by the expression of surface activation markers while impairing their apoptotic cell death. Therefore, PAA might be associated with inflammation [42]. It also suppressed macrophage functions [43]. PAA increased reactive oxygen species (ROS) and tumor necrosis factor (TNF)α release in vascular endothelial cells, which are known to contribute to atherosclerosis and blood vessel calcification [44]. PAA inhibited osteoblast proliferation [45]. It diminished inducible nitric oxide synthase (iNOS) expression in mononuclear leukocytes—nitric oxide (NO) released by iNOS activity is known to be protective in atherosclerosis [46].

Another uremic toxin is quinolinic acid (QA), which is increased in uremia [47] and likely responsible for diverse pathological features such as the development of anemia by antagonizing erythropoietin release, mediating some immunosuppression, neurotoxicity, and cardiovascular effects [38,47,48,49].

However, so far, nothing is known about the direct effect of these uremic toxins on tendon metabolism and their possible interaction with fluoroquinolones. Serum and plasma concentrations of QA and PAA are known to be substantially increased in uremia compared to normal conditions (Table 1), but concentrations in the bradytrophic tissue tendon are so far unknown. Knowledge of the shared mechanisms of fluoroquinolones and uremic toxins might allow tendon ruptures to be obviated and tendon repair strategies to be improved in hemodialysis patients.

### 2.1. Effects of Ciprofloxacin and Uremic Toxins on Tenocyte Metabolic Activity

CPX, when administered at higher concentrations (100 mg/L), significantly suppressed tenocyte metabolic activity after short-term exposure (8 h; Appendix A) and, in addition, after prolonged exposure (72 h) at therapeutic concentrations (Figure 2). PAA significantly reduced tenocyte metabolic activity only after prolonged exposure (72 h) either to very high doses (10 mM) or when combined with CPX (already with the lowest tested concentration of 3.5 mM; Figure 2). However, the suppressive effect of the combined treatment (PAA + CPX) did not significantly differ from that induced by CPX alone.

QA treatment led to no significant cytotoxic effect on cultured tenocytes after 72 h, even at the highest concentration tested (50 mg/L). The combinations of CPX and QA did not cause greater cytotoxicity than incubation with CPX alone (Figure 2).

### 2.2. Tenocyte Survival

In addition to measuring metabolic activity in response to CPX and uremic toxins, viability staining was performed to visualize viable and dead cells and to calculate their numbers after 72 h of exposure. The mean percentages of dead cells were below 20% in all treatment courses. There was only a tendency of increasing numbers of dead cells in response to CPX alone detectable. However, when administered at higher concentrations (10 mM), PAA elevated the number of dead cells significantly. Light microscopy observation revealed some cell clusters and cell loss in cultures treated with PAA combined with CPX (Appendix A). In combination with 3 mg/L CPX, after treatment with 3.5 mM PAA, a significant increase in the number of dead cells was already detectable. QA had no significant suppressive effect on tenocyte survival in the absence of CPX, but the combination of 3.3 mg/L QA with 10 mg/L CPX significantly increased the amount of cell death (Figure 3 and Figure 4A–C).

### 2.3. Effects of Ciprofloxacin and Uremic Toxins on Tenocyte Gene Expression

CPX, PAA, or QA had no significant effect on the β1-integrin (*ITGB1*) expression. Only the combination of CPX (3 mg/L) with PAA (3.5 mM) led to a significant suppression of *ITGB1* gene expression (Figure 5A).

The *MMP-1* mRNA levels were significantly increased by CPX (10 and 30 mg/L). This effect became more pronounced in the form of a synergism following exposure to a combination of CPX and PAA (10 mg/L and 10 mM). QA and its tested combinations with CPX did not influence *MMP-1* mRNA levels significantly (Figure 5B).

Gene expression of the pro-inflammatory cytokine *IL1B* was reduced by CPX after 72 h compared with the untreated controls. By contrast, PAA and QA significantly up-regulated *IL1B*mRNA levels at the highest concentrations tested (10 mM PAA and 10 mg/L QA). Co-stimulation of the cells with 10 mM PAA and 10 mg/L CPX led to a significantly higher *IL1B* mRNA expression than the stimulation with CPX alone at the same concentration (Figure 5C).

### 2.4. Effects of Ciprofloxacin and Uremic Toxins on Tenocyte Collagen Synthesis, Cytoskeleton, and MMP-1

Protein levels of intra- and extracellular type I collagen, visualized by immunolabeling of tenocytes cultured on cover slips and treated with CPX, decreased in response to high CPX doses (10 and 30 mg/L), whereas PAA and QA—even in combination with CPX—did not affect collagen type I synthesis (Figure 6; Figure 7). However, the suppressive effect of CPX on tenocyte collagen synthesis visualized by immunolabeling did not reach the significance level, but Western blot analysis revealed a significant suppression of collagen type I synthesis in tenocytes exposed to 30 mg/L CPX for 72 h (Figure 8).

All tenocytes, irrespectively whether treated or not, formed multiple intracellular F-actin bundles (stress fibers) connected to the cells’ focal adhesion sites. The organization of the F-actin cytoskeleton and amount of stress fibers in tenocytes did not show major differences in response to the various treatment courses. However, cells exposed to PAA appeared to have a more slender morphology (Appendix A).

The effect of CPX on β1-integrin was investigated on the protein level by Western blot analysis and even a high concentration of 30 mg/L did not affect tenocyte β1-integrin expression.

MMP-1 was barely detectable intracellularly at the protein level; hence, no effect could be demonstrated by means of immunolabeling.

## 3. Discussion

Based on an increasing number of case reports, hemodialysis has been suggested to be a risk factor for tendon ruptures [32,33,35,54,55,56,57,58,59,60,61]. However, published reports concerning the direct impact of uremic toxins and a possible additive effect with CPX on tendon-derived cells were not available. Hence, the present in vitro study was performed to investigate the impact of two selected uremic toxins (PAA and QA) alone or combined with CPX on tenocytes. For selecting a reasonable concentration for stimulating the tenocytes in this study the serum concentrations have been considered. The peak serum concentrations of CPX, given orally or intravenously, range from 0.5 to 10 µg/mL [62] (Table 1). In the present study, higher concentrations were also tested, since the levels might be locally and systemically elevated in renal insufficiency patients. The plasma concentration of PAA in normal individuals is < 1.4 mg/L, while, during uremia, it can increase to 467.2 mg/L [46] (Table 1). QA normally has a serum concentration of 0.1 mg/L, but can increase during uremia to 1.5 mg/L and can reach a maximum of 3.3 mg/L [38] (Table 1). PAA is partly (30%) protein-bound and, hence, difficult to eliminate by hemodialysis [39]. It is an inhibitor of Ca^++^-ATPase [46]. QA is a degradation product of the essential amino acid tryptophan, which is cleaved at higher rates under the conditions of renal insufficiency and likely contributes to immunosuppression, cardiovascular complications and neurotoxic side effects [38,48,63,64,65]. Tenocytes harvested from 14 healthy and young donors (mean age 34.9 years, 50% male) of hamstring tendons were tested in the present study for the effect of PAA, QA, and CPX. It has to be considered that, possibly, tenocytes from Achilles tendons, the tendon most often ruptured in response to CPX, might be more sensitive. In addition, tenocytes from patients with renal insufficiency and older ages might differ in their response, since both conditions predispose tendon tissue to rupture and, hence, the tendon degeneration preceding it [12,30]. In the present study no donor dependency was observed probably due to the small donor collective included.

When using these cells from healthy tendons, after 72 h at all concentrations tested, CPX exerted a significant suppressive effect on tenocyte metabolic activity. PAA impaired it at the same time only at higher concentrations, with no detected additional enhancement by CPX in co-treated cells, while QA had no significant effect. The measured impaired rates of metabolic activity could result from an inhibition of cell proliferation, reduced cellular activity or from increased rates of cell death, which have already been described for CPX [19,66]. Hence, the rate of dead cells was determined.

The rate of dead cells in CPX-treated tenocytes was higher (around 5%) than in the controls with only 0.8% of dead cells in the present study but remained low at all. In accordance with the above-mentioned suppression of tenocyte metabolism, PAA significantly induced tenocyte death. Interestingly, QA at its highest concentration, when combined with CPX, increased the rate of dying cells. Further studies should examine whether apoptosis or necrosis occurs, and which cell death pathways are most likely to be activated. Since tendon is a hypocellular tissue and few tenocytes have to regulate ECM de novo synthesis and remodeling the loss of few tenocytes might substantially impair tendon`s adaptability to tension and strength.

Tendon ruptures as observed in response to CPX treatment or renal insufficiency could be the result of a loss of tendon stability due to dysbalanced ECM remodeling or cell–ECM interaction. Type I collagen, the main tendon ECM protein, is a ligand for integrin receptors expressed on the tenocyte surface, particularly those possessing the β1-integrin chain. β1-integrins mediate the interaction between tenocytes and their surrounding ECM as a precondition for tendon integrity, but their function is divalent cation dependent [24]. Although, the suppressive effect of CPX (3–160 mg/L) on integrins has been shown in finger tendon-derived tenocytes and chondrocytes by others [24,66], to be explained by divalent cation deprivation mediated by the chelating properties of CPX, it could not be shown at the gene and protein expression level under the experimental setting of the present study. However, we did not address integrin activation in the present study. Integrins can be activated by inside-out signaling [67]. Under this condition cytoskeletal changes lead to integrin clustering in the cell membrane or conformational changes which influence their ligand binding affinity. The modulation of integrin glycosylation has also been implicated in integrin activation [67,68]. Hence, we checked the hyperglycosylated and less glycosylated forms of integrins separately in Western blot analysis, but we found, in agreement with the gene expression data, no regulation by CPX.

The effect of CPX on these important aspects is unknown; however, another class of quinolone compounds has been described as integrin antagonists [69] and CPX inhibited chondrocyte adhesion on collagen indicating disturbed integrin function [24].

The number of integrins as a correlate of gene and protein expression might determine function, such as enhanced cell adhesion in response to larger integrin clusters exposed on the cell membrane. Surprisingly, the gene expression of β1-integrin (*ITGB1*) was suppressed by a combination of CPX and PAA in the present study. This additive effect of CPX and PAA on β1-integrin gene expression is of interest since both agents alone had no effect. We can conclude that the same target of the same or crossing signaling pathways might be affected by CPX and PAA. One could study the Ca^++^ homeostasis, since PAA affects Ca^++^ ATPase [46], as mentioned above, and CPX can act as a chelator. F-actin visualization demonstrated stress fibers but revealed no differences of F-actin cytoskeleton between the treatment courses. Stress fibers are bundled at focal adhesion sites where usually integrin clusters are localized. Hence, the staining could indirectly visualize integrin-associated focal adhesion sites. Integrins are important mediators of tenocyte/ECM homeostasis [67,70].

To adapt their ECM to novel conditions, cells release MMPs such as MMP-1 [71]. Therefore, the expression of MMP-1 expression, which is the most important MMP in tendons and is capable of cleaving the mean component of the tendon ECM, type I collagen, was investigated.

*MMP-1* gene expression was induced by CPX, an observation supported by other studies [23,66]. However, in the study of Corps et al. [72], *MMP-1* gene expression was only substantially induced by poststimulation with IL-1β. This inductive effect of CPX was enhanced in the present study by a combination with PAA, but not by co-stimulation with QA. Nevertheless, distinct MMPs are intimately regulated by their activation through cleavage of proenzymes and by local inhibition by specific TIMPs [71]. This study is limited by the fact that the MMP activation and the balance between TIMPs and activated MMPs under the influence of PAA and QA was not addressed. These factors should be further investigated in cell culture supernatants in future.

The direct effect of CPX, PAA, and QA on the main ECM component type I collagen was investigated here, and neither PAA nor QA, either alone or in combination with CPX, was found to have a significant suppressive effect on its protein expression, as shown by image analysis, but a suppressive trend was detectable in response to 30 mg/mL CPX. Since cells were permeabilized for immunolabeling, the intracellular collagen type I precursor was shown here by the antibody localized mainly in the rough endoplasmatic reticulum (rER) region of the cells. The collagen released by the cells could not be visualized by this technique. As an important limitation of our present study, we did not quantify the release of collagen type I in the tenocyte culture supernatants. Western blot analysis revealed for the same treatment by CPX a significant collagen type I suppression. This observation of limited effects on tenocyte collagen expression is supported by the study of Chang et al., (2012), who found no suppression of type I collagen by CPX but an increase in type III collagen. Menon et al. detected no effect neither on type I nor III collagen mRNA and protein expressions [22]. This collagen type, which is known to increase in tendons during repair and remodeling [73], was not tested here. The presence of substantial collagen type I remodeling has been abrogated in the core of mature healthy tendons, as shown by two important studies of Heinemeier et al., [74,75]. However, the authors hypothesized that collagen turnover precedes the onset of tendinopathy [75].

IL-1β is a typical proinflammatory cytokine known to play a role in tendon disorders such as tendinopathy [76]. Interestingly, IL-1β (*IL1B*) gene expression was reduced by CPX in the present study. Despite the fact that there exists no comparable data in published studies; results obtained by others indicate an antagonism between CPX and IL-1β, for example, by reducing IL-1β induced PGE2 release by CPX [77]. In contrast to CPX, 10 mg/L PAA significantly induced gene transcription of this important proinflammatory cytokine, even in the presence of CPX, suggesting that it might contribute to the development of tendinopathy. The present study is limited by testing only two uremic toxins separately in a 2D model, since multiple interactions might occur among many uremic compounds accumulated in hemodialysis patients, possibly being involved in predisposing their tendons to rupture.

## 4. Materials and Methods

### 4.1. Preparation of Agents Used for Tenocyte Stimulation

The quinolone ciprofloxacin (CPX) was obtained from commercially available solutions for intravenous infusion (CPX-Infusionslösung, 200 mg/100 mL (Fresenius Kabi, Bad Homburg vor der Höhe, Germany). The uremic toxins phenylacetic acid (PAA) and quinolinic acid (QA) were purchased from Sigma-Aldrich (Taufkirchen, Germany). Concentrated stock solutions were prepared. PAA was diluted in ethanol (stock solution, 500 mM, pH 7.0) with sonication and further diluted to 100 mM in distilled water (pH 7.0, sterile filtered). A sterile (pyrogen-free) NaCl solution (0.9% *w*/*v*) was used as a solvent for QA (500 mM and 100 mM stock solutions, sterile filtered). The pH was adjusted to 7.0. Stock solutions were stored at −20 °C. All further dilutions were done with cell culture medium.

### 4.2. Tenocyte Isolation and Culture

Human primary tenocytes were isolated from the midsubstance of 14 hamstring tendons (*Musculus (M). semitendinosus, M. semimembranosus, M. gracilis* tendons) of healthy middle-aged donors (average age 34.9 years; 7 females, 7 males). This study was approved by the Charité review board (EA4-033-08, 22 May 2008) for experiments with human-derived tissues. The surrounding connective tissue and peritendineum of the tendons was carefully removed before culturing 2 × 5 mm tendon explants in a growth medium for several days. After 7–14 days, tenocytes continuously emigrated from these explants and adhered to petri dishes. When cell density approached confluence, these cells were detached using 0.05% trypsin/1.0 mM EDTA (Biochrom, Berlin, Germany) and further expanded in T75 flasks. Cells were cultured at 37 °C in a humidified atmosphere with 5% CO_2_. The growth medium was composed of Ham’s F-12/Dulbecco’s modified Eagle’s medium (50/50, Biochrom) containing 10% fetal calf serum (FCS, Biochrom), 25 μg/mL ascorbic acid [Sigma-Aldrich], 50 IU/mL streptomycin, 50 IU/mL penicillin, 2.5 μg/mL amphotericin B, essential amino acids, L-glutamine (all: Biochrom), and it was changed every three days. For the experiments, tenocytes were cultured at 15,625 cells/cm^2^ in 6-, 12-, or 96-well plates (Sarstedt AG, Nümbrecht, Germany) in the growth medium with 1% FCS. To achieve enough cells for all experiments tenocytes were expanded at least until passage (P) 4-6. Most of the experiments were performed with cells of P4. We confirmed the expression of tendon-associated proteins such as scleraxis, mohawk, tenomodulin, tenascin C, decorin, and CD90, typical for differentiated tenocytes in representative samples (Appendix A).

### 4.3. Tenocyte Stimulation with Ciprofloxacin and Uremic Toxins

To monitor the effects of CPX and uremic toxins, tenocytes were stimulated with the respective agents, alone or in combination. Tenocytes were serum-starved in a growth medium containing 1% FCS for 24 h before being incubated with the agents at different concentrations (CPX—3, 10, 30, 50, and 100 mg/L; PAA—3.5, 5, 10, 30 and 50 mM; QA—1.5, 3.3, 10, 50 and 100 mg/L). Since the solvent did not influence the results, as shown by preliminary experiments, it was omitted later in the control experiments. Hence, the control group was treated with a serum-reduced medium in the absence of stimulating agents.

### 4.4. AlamarBlue Assay

The cytotoxicity of the agents was determined by analyzing human tenocytes for their metabolic activity in response to treatment with CPX and uremic toxins. Tenocytes were cultured in 96-well plates at 15,625 cells/cm^2^ (Sarstedt AG, Nümbrecht, Germany). A 10% alamarBlue assay reagent (Life Technologies, Carlsbad, CA, USA) was added to the culture medium either immediately together with the stimulation agents (CPX, PAA, QA) or after the cells had been incubated with these agents for 72 h. The effect on metabolic activity in response to the different treatments was compared to the untreated control (culture medium with cells but without test reagent), which represented 100% metabolic activity. Induction of cell death was performed using sodium dodecyl sulfate (SDS; Carl Roth GmbH, Karlsruhe, Germany) as a negative (toxic) control at a concentration of 1 mM. The experiments were carried out independently using tenocytes of four different donors. The readout (2, 4, 6, 8, and 24 h) was performed via absorbance measurement at a wavelength of 570 nm and at a reference wavelength of 600 nm using a Genios microplate reader (Tecan Group AG, Männedorf, Switzerland).

### 4.5. Vitality Assay

Tenocytes seeded on Poly-L-lysin (Biochrom)-coated cover slides and stimulated with the agents for 72 h were incubated in a mixture of 5 µL/mL fluorescein diacetate (Sigma-Aldrich, 3 mg/mL dissolved in acetone (stock solution)) and 1 µg/mL ethidium bromide (Carl-Roth, Karlsruhe, Germany) diluted in 1 mL phosphate-buffered saline (PBS) for 10 min. The green (vital) or red (dead) cell fluorescence was visualized using fluorescence microscopy (Axioskop 40, Carl Zeiss, Jena, Germany) using a digital camera (Color View II, Olympus, Shinjuku, Japan). Three microscopic fields of each treatment group in each independent experiment were evaluated using ImageJ for vital and dead cells.

### 4.6. Gene Expression Analysis

Real-time detection polymerase chain reaction (RTD-PCR) analysis was performed to obtain semiquantitative gene expression data for β1-integrin (*ITG1*), *MMP-1*, and IL-1β (*IL1B*) versus hypoxanthine phosphoribosyltransferase (HPRT) as a reference gene. Tenocytes were cultured at 15,625 cells/cm^2^ in 6-well plates for 24 h before stimulation. The cells were then rinsed with PBS, serum-starved (growth medium with 1% FCS) for 1 h, and stimulated for 72 h with the agents. The tenocytes’ total RNA was isolated using a Qiagen RNA isolation mini kit (Qiagen, Hilden, Germany), and the RNA quantity and purity were evaluated using the RNA 6000 Nano assay (Agilent Technologies, Santa Clara, CA, USA). Reverse transcription was performed using the Quanti Tect Reverse Transcription Kit and equal amounts of RNA (500 ng) according to the manufacturer’s instructions (Qiagen). Aliquots of 1 µL cDNA (16.7 ng) of each sample were amplified using RTD-PCR in a 20 µL reaction using the TaqMan Gene Expression Assay (Applied Biosystems (ABI), Foster City, CA, USA) and specific primer pairs (see Table 2, ABI) using an Opticon 1 Real-Time-Cycler (Opticon^TM^ RTD-PCR, Biorad, Hercules, CA, USA). The following conditions of amplification were chosen—2 min at 50 °C, 10 min at 95 °C (for Uracil-N-glycosylase activation and denaturation to remove carry overs), and then, for 40 cycles, 15 s at 95 °C, 60 s at 60 °C, and then 6 °C cooling. Relative gene expression levels were normalized and calculated using the 2^−ΔCT^ method [78].

### 4.7. Immunofluorescence Microscopical Analysis

Tenocytes were cultured on cover slips for 24 h, serum-starved for 1 h, and treated for 72 h with the different agents before fixing in a 4% paraformaldehyde ready-to-use solution (PFA, Carl Roth, Germany) for 15 min. Tenocytes were washed with Tris-buffered saline (TBS—0.05 M Tris, 0.15 M NaCl, pH 7.6) and incubated with protease-free donkey serum (5%, diluted in TBS and 0.1% Triton X-100) for 20 min at ambient temperature (RT). Subsequently, cells were rinsed and incubated with the primary antibodies in a humid chamber for 24 h overnight at 4 °C. The following primary antibodies were used—rabbit anti-human type I collagen (Acris Antibodies, Hiddenhausen, Germany), mouse anti-human MMP-1 (R&D Systems, Minneapolis, MN, USA), rabbit anti-human mohawk (Biozol, Eching, Germany), scleraxis, decorin (both: Acris Antibodies); mouse anti-human tenascin C (GeneTex Inc. Biozol), CD90 (Miltenyi Biotec, Bergisch Gladbach, Germany); goat anti-human tenomodulin (Santa Cruz Biotechnology, Dallas, TX, USA), as well as mouse and rabbit isotype IG1 antibodies (BD Bioscience, Franklin Lakes, NJ, USA, and Invitrogen, Carlsbad, CA, USA). Tenocytes were subsequently washed three times (each 5 min) with TBS before incubation with donkey anti-mouse or -rabbit Alexa-Fluor^®^488 (Invitrogen), donkey anti-rabbit Alexa-Fluor^®^555, donkey anti-mouse or -goat cyanine 3-coupled secondary antibodies or Phalloidin-Alexa488 (diluted 1:200 in TBS), and 4′,6′-diamidino-2-phenylindol (DAPI) (Roche Diagnostics, Rotkreuz, Switzerland) for counterstaining cell nuclei, respectively, for 1 h at RT. Immunolabeled tenocytes were washed three times with TBS, before being mounted using a Fluoromount mounting medium (Southern Biotech, Biozol Diagnostica, Eching, Germany) and examined under a fluorescence microscope (Axioskop 40, Carl Zeiss, Jena, Germany). The immunofluorescence images were taken under standardized conditions at 200× magnification. Three images were taken for each staining, and, in each image, four to eight cells were analyzed for fluorescence intensity using the ImageJ software (Rasband, National Institutes of Health, Bethesda, MD, USA). According to the average fluorescence intensity value, the background fluorescence was subtracted.

### 4.8. Western Blot Analysis and Densitometric Evaluation

Type I collagen expression was analyzed using Western blot analysis using β-actin as the reference protein. Tenocytes were rinsed twice with ice-cold PBS and then incubated with a 0.3 mL lysis buffer (1 tablet of complete Mini Protease Inhibitor Cocktail [Roche Diagnosticsnd], 100 µL 200 mM 1,4-dithio-DL-threitol (DTT; Carl Roth), 100 µL 0.1 M ethylene glycol tetra acetic acid (EGTA; Carl Roth), 250 µL 1 M 4-(2-hydroxyethyl)-1 piperazine ethanesulfonic acid (HEPES; Biochrom), 100 µL 0.5 M magnesium chloride (MgCl_2_; Sigma-Aldrich), 100 µL 10% Triton X-100 (Sigma-Aldrich), and ultrapure water ad 10 mL for 5 min. Cell disintegration was achieved by scraping of the adherent cells using cell scrapers (TPP, Trasadingen, Switzerland). The cell lysate was pipetted into 1.5 mL tubes and centrifuged at 17,000× *g* and 4 °C for 30 min. The supernatants containing total cell proteins were transferred into novel 1.5 mL reaction tubes and mixed with 6× Laemmli buffer (1 M Tris HCl pH 6.8, 4.2% sodium dodecyl sulfate (SDS), 2.1% β-mercaptoethanol (Sigma-Aldrich), bromophenol blue (Serva, Heidelberg, Germany), 7.5% glycerine (Sigma-Aldrich)) to reach a target volume of 100 µL and denaturized by incubation at 95 °C for 10 min. Protein samples (20 µL per lane) were separated by SDS polyacrylamide gel electrophoresis (SDS-PAGE) at 80 V for 30 min and 120 V for 60–90 min using 7.5% separating and 5% collecting gels. Ten to 15 µL Precision Plus Protein™ Kaleidoscope Standards (Bio-Rad Laboratories, Munich, Germany) loaded into the first slot served as a protein size reference. A polyvinylidenfluoride (PVDF) membrane (0.45 µm pore size (Merck Millipore, Burlington, MA, USA)) was immersed for 2 min in 100% methanol (Merck Millipore, Darmstadt, Germany) for activation before proteins were transferred onto it using 120 V for 90 min. Thereafter, the PVDF membrane was blocked with blocking buffer (1:9 Roti^®^-Block (Carl Roth) in distilled water) at room temperature (RT) for 2 h to saturate any unspecific binding sites. The membranes were incubated with the specific primary antibody overnight at 4 °C (rabbit anti-human collagen type I (Acris Antibodies, Hiddenhausen Germany)) or for 1 h at RT (β-actin, A5441, Sigma-Aldrich). The membranes were then washed once in PBS, two-times with washing buffer (0.5 mL Tween-20 [Sigma-Aldrich] in PBS ad 1.000 mL), and once again with PBS. Incubation with the enzyme-coupled secondary antibodies (goat anti-mouse and goat anti-rabbit (P0447 and P0448), DAKO Cytomation, Hamburg, Germany) was performed for 2 h at RT. After washing as described above, the chemiluminescence reaction was initiated using horseradish peroxidase substrate peroxide solution (Merck Millipore) and luminol reagent (Merck Millipore). The resultant chemiluminescence was detected using high-performance chemiluminescence films (GE Healthcare Limited, Chalfont St. Giles, UK). Densitometric evaluation of the Western blot analyses was performed using Bio 1D software (Vilber-Luormat, Eberhardzell, Germany).

### 4.9. Statistical Analysis

Statistical analysis was performed using GraphPad Prism 6 software (GraphPad Software Inc., San Diego, CA, USA). Results were normalized versus the non-stimulated control and analyzed using a one sample t-test. ANOVA, followed by a post hoc Bonferroni test, was used to compare different experimental groups. The GRUBBS’ test was applied to detect outliers. If applicable, the Kolmogorov–Smirnov and the Shapiro–Wilk tests were used to analyze the data for the presence of a Gaussian distribution. All data were expressed as the mean and standard deviation of the mean (mean ± SD). Differences between experimental groups and controls were considered significant at * *p* ≤ 0.05, ** *p* ≤ 0.01, *** *p* ≤ 0.005 and **** *p* ≤ 0.001.

## 5. Conclusions

The results of this study could visualize distinct suppressive effects of the two selected uremic toxins, QA and, more pronouncedly, PAA, on tenocytes. PAA suppressed tenocyte metabolic activity and increased the rate of cell death, while QA induced *IL1B* gene expression at a concentration of 10 mg/L. The results also confirmed some of the previously reported effects of CPX on tenocytes suppressing metabolic activity and at high concentration collagen type I protein as well as induction of *MMP-1* gene expression. A possible additive effect with CPX could only be shown for PA, namely, detectable as a suppressed β1-integrin (*ITG1*) gene expression by tenocytes. However, there might exist multiple interactions with other important uremic toxins, such as β2-microglobulin, parathormone, neopterin, *p*-cresyl sulfate, or others, which should be analyzed in future studies.

## Figures and Tables

**Figure 1 ijms-21-04241-f001:**
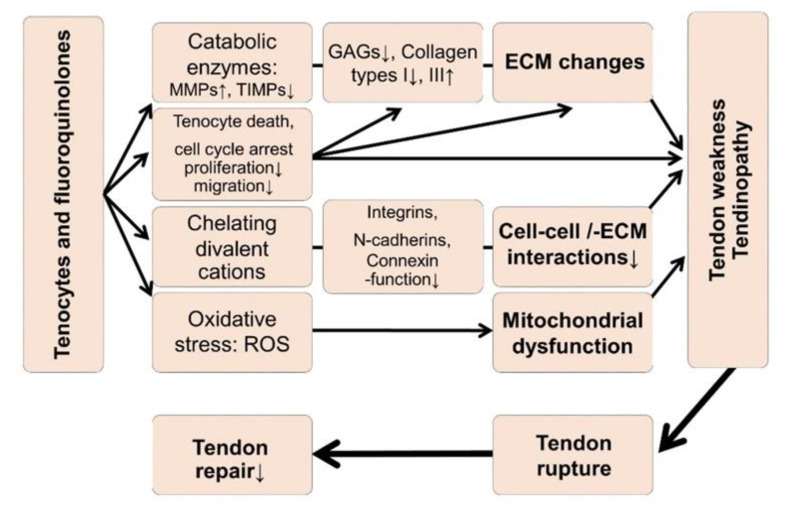
Hypotheses regarding tendinopathy and tendon rupture induced by fluoroquinolones. ECM: Extracellular tendon matrix, MMP: Matrix metalloproteinases, ROS: Reactive oxygen species, TIMP: Tissue inhibitor of metalloproteinases. The graph refers to [15,16,17,19,20,21,22,23,25,26,27,28,50].

**Figure 2 ijms-21-04241-f002:**
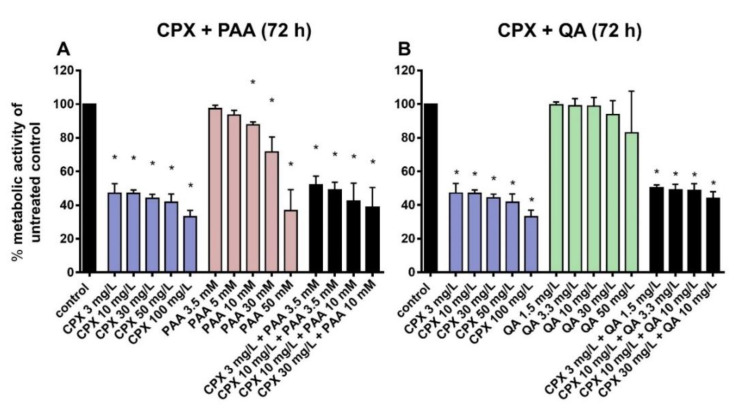
Metabolic activity after 72 h preincubation with CPX (**A**,**B**), PAA (**A**), QA (**B**) alone, and combinations of CPX and PAA (**A**) and CPX and QA (**B**). The effects of phenylacetic acid (PAA) and quinolinic acid (QA) alone or in combination with ciprofloxacin (CPX) on tenocyte metabolic activity were detected by measuring reduction of alamarBlue after 72 h of incubation. Results are expressed as a percentage of the activity observed in untreated controls. Bars show the mean ± standard deviation (SD) obtained from four independent experiments with tenocytes from different patients. * *p* ≤ 0.05 compared to control.

**Figure 3 ijms-21-04241-f003:**
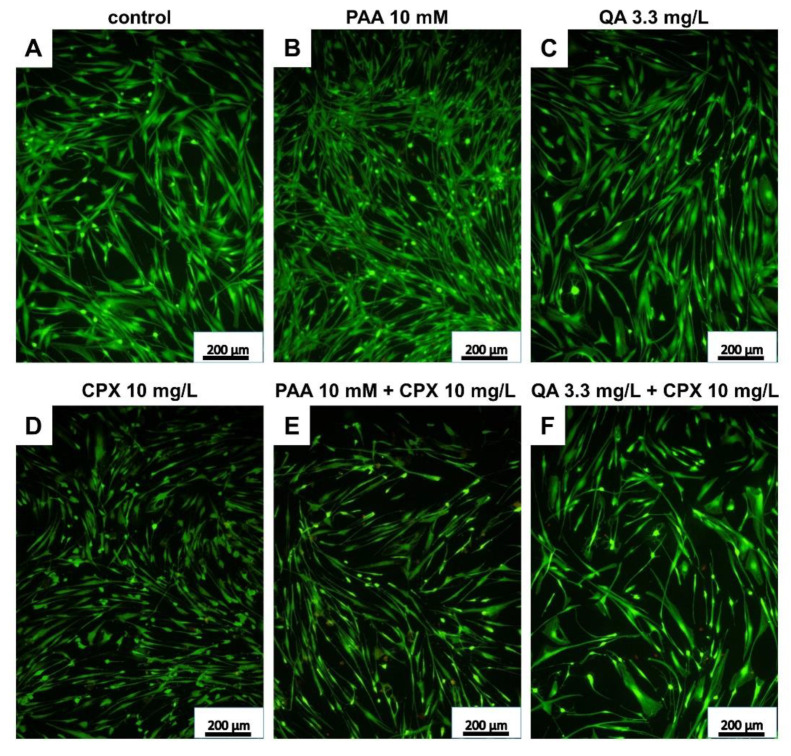
Live/dead staining of human tenocytes after treatment with PAA, QA, CPX, alone or in combination with CPX, for 72 h. Representative images are shown. (**A**) Untreated control, (**B**) PAA 10 mM, (**C**) QA 3.3 mg/L, (**D**) CPX 10 mg/L, (**E**) PAA 10 mM + 10 mg/L CPX, (**F**) QA 3.3 mg/L + CPX 10 mg/L. Living cells—green, dead cells—red. Scale bars = 200 µm.

**Figure 4 ijms-21-04241-f004:**
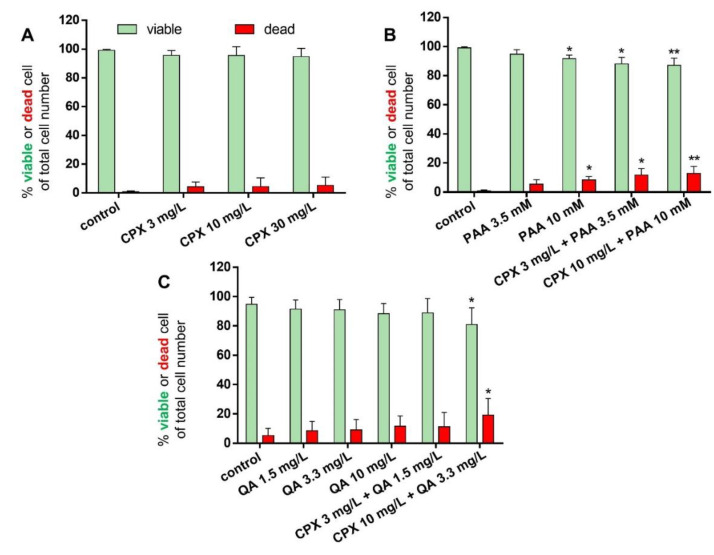
Percentage of surviving and dead tenocytes after 72 h incubation with CPX alone and with PAA and QA, alone and in combination with CPX. Percentages of viable (green bars) and dead cells (red bars) are shown after incubation with CPX alone for 72 h (**A**), PAA alone or combined with CPX (**B**), and QA alone or combined with CPX (**C**). Mean values ± standard deviation are depicted. Results derive from four to seven independent experiments with tenocytes of four to seven different donors. * *p* ≤ 0.05, ** *p* ≤ 0.01 compared to control.

**Figure 5 ijms-21-04241-f005:**
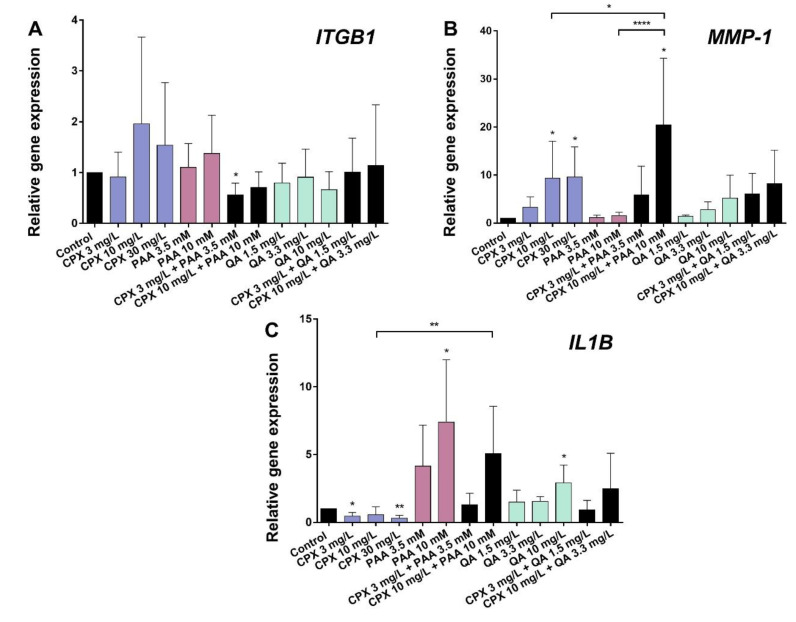
Gene expression of β_1_-integrin (*ITGB1*), *MMP-1*, and IL-1β (*IL1B*) in human tenocytes after 72 h incubation with CPX alone and with PAA and QA, alone and in combination with CPX. (**A**) *ITGB1*, (**B**) *MMP-1*, and (**C**) *IL1B*. Gene expression was normalized using the reference-gene hypoxanthine phosphoribosyltransferase (*HPRT*) and compared to the untreated control. Bars show the mean ± SD (five to six independent experiments with cells of five to six different donors were included). * *p* ≤ 0.05, ** *p* ≤ 0.01, **** *p* ≤ 0.001 compared to control.

**Figure 6 ijms-21-04241-f006:**
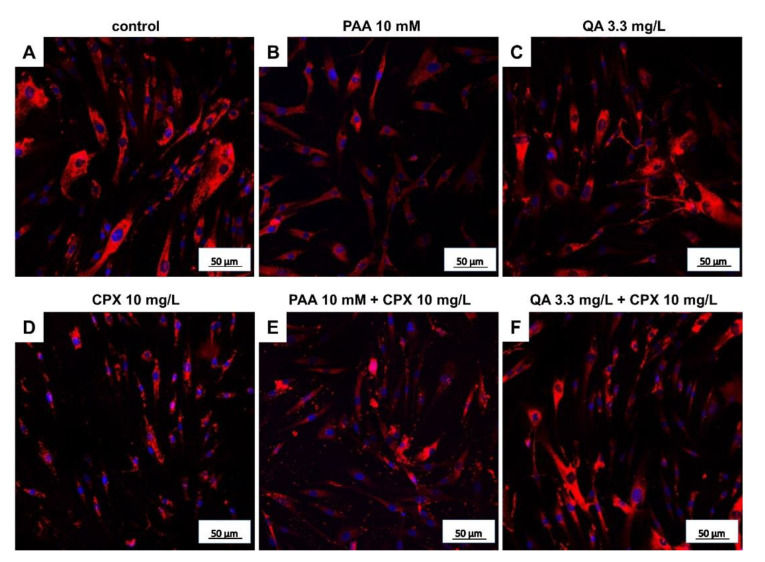
Tenocyte type I collagen expression after 72 h in response to stimulation with CPX alone and with PAA and QA, alone and in combination with CPX. Representative images of collagen type I immunolabeling (red) are shown. Cell nuclei are counterstained in blue using 4′,6′-diamidino-2-phenylindol (DAPI). (**A**) Untreated control, (**B**) PAA 10 mM, (**C**) QA 3.3 mg/L, (**D**) CPX 10 mg/L, (**E**) PAA 10 mM + 10 mg/L CPX, (**F**) QA 3.3 mg/L + CPX 10 mg/L. Scale bars = 50 µm.

**Figure 7 ijms-21-04241-f007:**
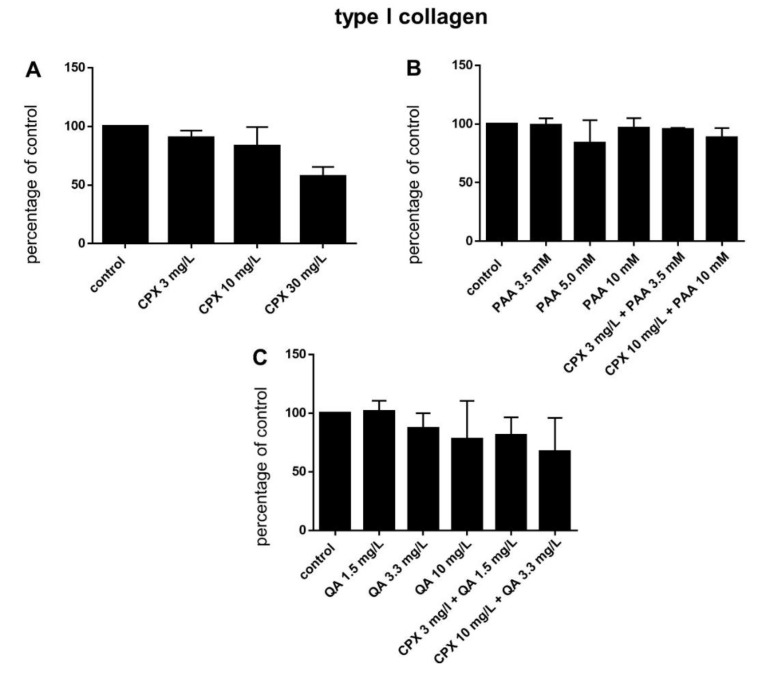
Tenocyte type I collagen expression after 72 h in response to stimulation with CPX alone and with PAA and QA, alone and in combination with CPX. (**A**) CPX, (**B**) PAA combined with CPX**,** (**C**) QA combined with CPX. Fluorescence intensities were analyzed using ImageJ. Bars show the mean ± SD.

**Figure 8 ijms-21-04241-f008:**
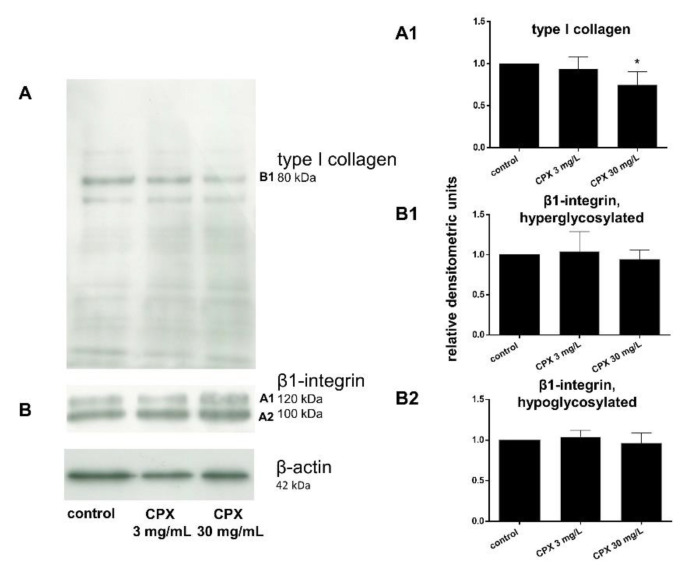
Type I collagen and β1-integrin expression of tenocytes treated with ciprofloxacin as shown by Western blot analysis and densitometric evaluation after 72 h. (**A**) Representative Western blot of type I collagen expression, (**A1**) densitometric evaluation of the type I collagen Western blots. (**B**) Representative Western blot of β1-integrin expression. The two visible bands (hyperglycosylated and hypoglycosylated forms (**B1**,**A2**)) of the β1-integrin were densitometrically evaluated in the Western blots performed for β1-integrin detection. Collagen type I, *n* = 4: β1-integrin, *n* = 5: Collagen type I. Independent experiments were performed with cells from different donors. ** p* ≤ 0.05 compared to control.

**Table 1 ijms-21-04241-t001:** Serum/plasma concentration in normal patients (CN) in comparison to mean concentrations in uremic patients (CU) and highest concentrations (CM).

	MW (g/mol)	C_N_	C_U_	C_M_	Unit	Reference
**Quinolinic acid**	167	0.1	1.5	3.3	mg/L	[37]
**Phenylacetic acid**	136	<1.4	467.2	474.6	mg/L	[46]
**Ciprofloxacin**	331	1.35 to 4.21	3.7	7.6	mg/L	[51,52,53]

**Table 2 ijms-21-04241-t002:** Primers used for RTD PCR in the present study (purchased via ABI, USA).

Gene Name	Amplicon Size	Assay ID
*HPRT1*	100 bp	Hs99999909_m1
*IL1B*	94 bp	Hs00174097_m1
*ITGB1*	75 bp	Hs00559595_m1
*MMP-1*	133 bp	Hs00233958_m1

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
