# Peer review of "Uremic Toxins and Ciprofloxacin Affect Human Tenocytes In Vitro"

_ijms, 2020, doi:10.3390/ijms21124241_

Round 1

Reviewer 1 Report

In this paper the authors explored the effects of the uremic toxins phenylacetic acid (PAA) and quinolinic acid (QA), both alone and in combination with ciprofloxacin (CPX), on human tenocytes in vitro, in order to find any synergistic tenotoxic effect.

The paper needs extensive modifications to improve especially the experimental design and the discussion. Moreover, in the present form the conclusions are not completely supported by the results. Finally, the discussion is a list of sentences, and a real discussion of the results is not provided at all.

Specific comments:

Results, page 6: the authors describe the results relative to MMP-1 gene expression. MMP-1 is the key MMP involved in type I collagen degradation and its activity is played in the extracellular matrix since it’s a secreted protein. Moreover, its activity and activation are under control of TIMPs. As a consequence, MMP-1 gene expression is not predictive, and MMP-1 should be assessed by Western blot or zymography, together with the analysis of TIMPs. For the same reason, the intracellular analysis of MMP-1 is not useful.

Results: type I collagen is a secreted protein undergoing remodeling by MMPs. Its quantification should be analyzed in the cell culture supernatant by molecular analysis.

Results: micrographs showing immunofluorescence results as well as Western blot should be provided.

Results: did the authors find any influence of donor age on the analyzed parameters?

Discussion, page 8, lines 146-146: please discuss the lacking additive effect of CPX and PAA.

Discussion, page 8, lines 157-162: this paragraph should be inserted at the beginning of the discussion.

Discussion, page 8: please discuss the possible effect of integrin downregulation.

Materials and Methods: at what passage tenocytes were used? what portion of the tendon was used to obtain tenocytes? this information should be included. Moreover, how many fields were counted to assess cell viability?

Author Response

Reviewer 1

Comments and Suggestions for Authors

The paper needs extensive modifications to improve especially the experimental design and the discussion. Moreover, in the present form the conclusions are not completely supported by the results.

Finally, the discussion is a list of sentences, and a real discussion of the results is not provided at all.

Response: The discussion section has substantially been restructured now. We elaborated the discussion of own results. We found indeed that the conclusion section contained some mistakes and corrected them.

Specific comments:

Results, page 6: the authors describe the results relative to MMP-1 gene expression. MMP-1 is the key MMP involved in type I collagen degradation and its activity is played in the extracellular matrix since it’s a secreted protein. Moreover, its activity and activation are under control of TIMPs. As a consequence, MMP-1 gene expression is not predictive, and MMP-1 should be assessed by Western blot or zymography, together with the analysis of TIMPs. For the same reason, the intracellular analysis of MMP-1 is not useful.

Response: We performed MMP-1 immunostaining previously but only few cells were positive, hence, semiquantification was not possible. We discuss the importance of the balance between MMPs and TIMPs in the discussion section now. We included the lacking investigation of TIMPs as a limitation of the present study in the discussion section. We added also the critisism of the reviewer in this context to the discussion section. We were not able to perform westernblot analysis and zymography for MMPs/TIMPs due to the situation of limited laboratory accessablity due to shut down. In addition, we did not have sufficient Hamstring tendon-derived tenocytes and novel cell isolation was not possible.

Results: type I collagen is a secreted protein undergoing remodeling by MMPs. Its quantification should be analyzed in the cell culture supernatant by molecular analysis.

Response: We did not store culture supernatants. Hence, we have to repeat the experiments, which was not possible at the moment for the above mentioned reasons. Instead, we show the collagen type I immunolabeling in the main body of the manuscript now which was intra- and extracellularly labeled and semiquantified using imageJ analysis (novel Figures 6 and 7).

Results: micrographs showing immunofluorescence results as well as Western blot should be provided.

Response: We show western blots for the collagen type I expression in response to CPX in the main body of the manuscript (Figure 8). Immunofluorescence images (collagen type I, intra- and extracellularly immunolabeled) of the whole experimental setting are shown in Figure 6 and 7. We decided not to focus so much on collagen type I since reviewer 2 argues that collagen type I remodeling plays only a minor role in native mature tendons underlined by experiments of Heinemeier et al., (2013; 2018). This literature is now cited and discussed in the discussion section.

Results: did the authors find any influence of donor age on the analyzed parameters?

Response: We investigated only a small cohort of healthy patients (14) of nearly similar age. Hence, we could not find any influence. We added this observation of no influence to the discussion section.

Discussion, page 8, lines 146-146: please discuss the lacking additive effect of CPX and PAA.

Response: We discussed it now. Possibly, both agents act concentration-dependent on the same signalling pathway e.g. by interfering with Ca++ dependent mechanisms (Fig. 1 in the manuscript and Jankowski et al., 2003.

Discussion, page 8, lines 157-162: this paragraph should be inserted at the beginning of the discussion.

Response: Thanks a lot for the valuable advice. We started the discussion section now with two introducing sentences followed by the sequence transposed as suggested by the reviewer.

Discussion, page 8: please discuss the possible effect of integrin downregulation.

Response: We discuss the integrin regulation and activation more thoroughly now together with our own data. Since activated integrins form clusters localized at focal adhesions sites (FAS) we decided to stain F-actin fibers („stress fibers“) which form bundles at the FAS (supplemental Fig. 3) and hence, indirectly show the presence of numerous FASs

Materials and Methods: at what passage tenocytes were used? what portion of the tendon was used to obtain tenocytes? this information should be included.

Response: We added the range of passage numbers (passages 4-6) in section 4.1: “To achieve enough cells for all experiments tenocytes were expanded at least until passage 4-6. Most of the experiments were performed with cells of P4.

We confirmed the expression of tendon-associated proteins such as scleraxis, Mohawk, tenomodulin, tenascin C, decorin and CD90, typical for differentiated tenocytes in representative samples (supplemental figure 4).” It is stated now in 4.1. that the midsubstance was used for cell isolation.

Moreover, how many fields were counted to assess cell viability?

Response: Three microscopic fields were analyzed. It is added in the method section now (4.4).

Reviewer 2 Report

The authors have addressed what antibiotic ciprofloxacin does for the metabolism of tenocytes in vitro. As tendon ruptures caused by ciprofloxacin are common in old people with chronic kidney disease, they have also quantified the effect of ciprofloxacin in the presence or the absence of uremic toxins. The experiments have been carefully carried out. I have no concerns related to the actual execution of the experiments, but my main point of concerns relates to using gene expression levels as the main outcome measure, especially the synthesis of tendon extracellular matrix (ECM). Namely, (14)C bomb-pulse method method has shed new light on the pathogenesis of tendon disorders. The quantification of 14(C) has demonstrated that no tissue renewal tissue takes place after the teenage years (roughly after age 17) in healthy tendon (Heinemeier et al. 2013, 2018).

As these results are probably true and explain issues such as the overall poor prognosis in tendon disorders, the measurements such as collagen gene expression might not be most relevant outcome measures. Thus, I want the authors to discuss their results in light of the results obtained from (14)C bomp pulse method.

Minor issues:

β1-integrin – The key point with integrin-mediated cell adhesion (especially in tendon, please see above) is not related to new integrin expression, but the activation state of integrin (low/intermediate/high), which defines how it interacts with surrounding ECM. I would assume that the integrin activation state is far more important for tenocyte survival than its expression levels. Please expand the discussion on integrins to include this scenario.

Suggested literature:

Heinemeier KM, Schjerling P, Heinemeier J, Magnusson SP, Kjaer M. Lack of tissue renewal in human adult Achilles tendon is revealed by nuclear bomb (14)C. FASEB J 2013 May;27(5):2074-2079.

(Heinemeier KM, Schjerling P, Ohlenschlaeger TF, Eismark C, Olsen J, Kjaer M. Carbon-14 bomb pulse dating shows that tendinopathy is preceded by years of abnormally high collagen turnover. FASEB J 2018 Mar 23:fj201701569R.

Author Response

Reviewer 2

I have no concerns related to the actual execution of the experiments, but my main point of concerns relates to using gene expression levels as the main outcome measure, especially the synthesis of tendon extracellular matrix (ECM).

Response: This critisism is discussed together with that of reviewer 1 and protein data are included in the discussion (collagen type I).

Namely, (14)C bomb-pulse method method has shed new light on the pathogenesis of tendon disorders. The quantification of 14(C) has demonstrated that no tissue renewal tissue takes place after the teenage years (roughly after age 17) in healthy tendon (Heinemeier et al. 2013, 2018).

Response: We thank the reviewer for the proposed interesting literature. We inserted and discussed it in the discussion section now (page 12, third paragraph).

As these results are probably true and explain issues such as the overall poor prognosis in tendon disorders, the measurements such as collagen gene expression might not be most relevant outcome measures. Thus, I want the authors to discuss their results in light of the results obtained from (14)C bomp pulse method.

Response: Done.

Minor issues:

β1-integrin – The key point with integrin-mediated cell adhesion (especially in tendon, please see above) is not related to new integrin expression, but the activation state of integrin (low/intermediate/high), which defines how it interacts with surrounding ECM. I would assume that the integrin activation state is far more important for tenocyte survival than its expression levels. Please expand the discussion on integrins to include this scenario.

Response: We discuss the integrin activation as an important aspect in the discussion section supported by respective literature. Together with the integrins we could discuss F-actin staining now since stress fibers indicate indirectly the presence of focal adhesion sites where integrins form usually clusters. It is shown as supplemental figure 3 now.

Suggested literature:

Heinemeier KM, Schjerling P, Heinemeier J, Magnusson SP, Kjaer M. Lack of tissue renewal in human adult Achilles tendon is revealed by nuclear bomb (14)C. FASEB J 2013 May;27(5):2074-2079.

(Heinemeier KM, Schjerling P, Ohlenschlaeger TF, Eismark C, Olsen J, Kjaer M. Carbon-14 bomb pulse dating shows that tendinopathy is preceded by years of abnormally high collagen turnover. FASEB J 2018 Mar 23:fj201701569R.

Round 2

Reviewer 1 Report

The authors addressed almost all issues and the revised manuscript was improved. However, some key issues were not addressed, in part due to the limited laboratory access due to shut down.

Major comments:

1- In the revised manuscript the authors reported that COL-I expression was detected by immunofluorecence both intra- and extracellularly: it is not possible to quantify “extracellular” collagen in cultured cells grown on coverlips. This sentence should be rephrased and the discussion modified accordingly.

2-Collagen turnover pathways must be characterized by analyzing collagen expression and MMP-1 activity in cell supernatants. Since this analysis was not performed, this represents an important limitation of the study. A paragraph specifically stating this limitation should inserted and the discussion should be smoothened in relation to this limitation.

Minor comments

Page 4, line 105: “Light microscopical observation” should be “Light microscopy observation”.

Author Response

Dear Editor,                                                                                                 5th June 2020

The authors would like to thank the reviewer for carefully his/her constructive comments. We modified the manuscript according to his/her suggestions. The changes performed are listed below point by point. They are indicated in red and underlined in the revised version of the manuscript. Please refer to our point by point reply below.

Sincerely,

Univ.-Prof. Dr. Gundula Schulze-Tanzil

The authors addressed almost all issues and the revised manuscript was improved. However, some key issues were not addressed, in part due to the limited laboratory access due to shut down.

Major comments:

1- In the revised manuscript the authors reported that COL-I expression was detected by immunofluorecence both intra- and extracellularly: it is not possible to quantify “extracellular” collagen in cultured cells grown on coverlips. This sentence should be rephrased and the discussion modified accordingly.

Response: p12, line 228-229 was changed now.

2-Collagen turnover pathways must be characterized by analyzing collagen expression and MMP-1 activity in cell supernatants. Since this analysis was not performed, this represents an important limitation of the study. A paragraph specifically stating this limitation should inserted and the discussion should be smoothened in relation to this limitation.

Response: we inserted this limitation of our study now (Lines 224-226, lines 231-234) and adapted the discussion accordingly.

 Minor comments

Page 4, line 105: “Light microscopical observation” should be “Light microscopy observation”.

Response: we changed the text accordingly.

Round 3

Reviewer 1 Report

The manuscript was greatly improved.